# Interactive Dairy Goat Image Segmentation for Precision Livestock Farming

**DOI:** 10.3390/ani13203250

**Published:** 2023-10-18

**Authors:** Lianyue Zhang, Gaoge Han, Yongliang Qiao, Liu Xu, Ling Chen, Jinglei Tang

**Affiliations:** 1College of Information Engineering, Northwest A&F University, Yangling, Xianyang 712100, China; zhangly@nwafu.edu.cn (L.Z.); hangaoge@nwafu.edu.cn (G.H.); xu_liu_work@163.com (L.X.);; 2Australian Institute for Machine Learning (AIML), The University of Adelaide, Adelaide 5005, Australia; yongliang.qiao@ieee.org; 3The Key Laboratory of Agricultural Internet of Things, Ministry of Agriculture, Yangling, Xianyang 712100, China; 4Shaanxi Key Laboratory of Agricultural Information Perception and Intelligent Service, Yangling, Xianyang 712100, China

**Keywords:** dairy goat, interactive segmentation, deepLabv3+, deep learning, precision stock farming

## Abstract

**Simple Summary:**

Deep-learning-based algorithms have achieved great success in intelligent dairy goat farming. However, these algorithms require a large load of image annotation to obtain a decent performance. The existing annotation of dairy goat images heavily relies on non-intelligent tools such as Labelme, which makes it extremely inefficient and time-consuming to obtain a high-quality annotation result, hindering the application and development of deep-learning algorithms in intelligent dairy goat farming. In this study, we explore an interactive annotation method based on deep-learning algorithms for dairy goat image annotation, which significantly reduces the annotation workload of the user. Specifically, it only takes 7.12 s on average to annotate a dairy goat image with our developed annotation tool, five times faster than Labelme, which takes an average of 36.34 s per instance.

**Abstract:**

Semantic segmentation and instance segmentation based on deep learning play a significant role in intelligent dairy goat farming. However, these algorithms require a large amount of pixel-level dairy goat image annotations for model training. At present, users mainly use Labelme for pixel-level annotation of images, which makes it quite inefficient and time-consuming to obtain a high-quality annotation result. To reduce the annotation workload of dairy goat images, we propose a novel interactive segmentation model called UA-MHFF-DeepLabv3+, which employs layer-by-layer multi-head feature fusion (MHFF) and upsampling attention (UA) to improve the segmentation accuracy of the DeepLabv3+ on object boundaries and small objects. Experimental results show that our proposed model achieved state-of-the-art segmentation accuracy on the validation set of DGImgs compared with four previous state-of-the-art interactive segmentation models, and obtained 1.87 and 4.11 on mNoC@85 and mNoC@90, which are significantly lower than the best performance of the previous models of 3 and 5. Furthermore, to promote the implementation of our proposed algorithm, we design and develop a dairy goat image-annotation system named DGAnnotation for pixel-level annotation of dairy goat images. After the test, we found that it just takes 7.12 s to annotate a dairy goat instance with our developed DGAnnotation, which is five times faster than Labelme.

## 1. Introduction

With the continuous development of the livestock industry, current dairy goat farming is gradually developing from previous retail farming to large-scale, intensive, and intelligent farming. Intelligent video surveillance plays a significant role in intelligent dairy goat farming [1,2,3], which is more real-time, active, and intelligent than traditional video surveillance. To be specific, traditional video surveillance focuses more on video recording rather than real-time monitoring and is mainly used by managers for after-the-fact review, which cannot understand the content in surveillance videos. In contrast, intelligent video surveillance is based on deep-learning algorithms, which can analyze and understand the content of the video, providing farmers with more valuable information such as dairy goat growth statistics and abnormal-behavior monitoring reports of dairy goat individuals [4], and enable them to make timely interventions in emergencies that may occur during dairy goat farming.

Notably, segmentation-based algorithms such as semantic segmentation and instance segmentation are the premise and basis for the analysis and understanding of dairy goat surveillance video content. For example, semantic segmentation of dairy goat farm images can realize dairy goat farm environment perception and then can be used as the eyes of robots to assist them in planning routes to clean up dairy goat farm garbage and add fodder to troughs, which can vastly improve the management efficiency of dairy goat farming and reduce the labor cost of dairy goat farming. In addition, instance segmentation of dairy goat individuals in the image can realize automatic estimation of dairy goat live body weight [5,6], and then realize the real-time monitoring of the dairy goat growth, and further improve the scientific and rational qualities of dairy goat farming [7].

However, semantic segmentation and instance segmentation require a large amount of pixel-level annotated images for model training. Currently, users mainly employ Labelme to annotate their own datasets. It requires users to draw a polygon that closely encloses the object along the boundary of the object by clicking, which is too inefficient and time-consuming to obtain a high-quality annotation result, as shown in Figure 1.

Compared with non-intelligent image-annotation tools (e.g., Labelme), interactive image annotation can significantly reduce the user’s annotation workload [8,9]. To illustrate this, for the annotation of the same dairy goat image, users can achieve the desired annotation result with only several clicks using the interactive annotation method, while the use of Labelme requires a significantly higher number of clicks (i.e., dozens of clicks and even higher), as evident in Figure 1 and Figure 2, respectively. However, in order to improve the efficiency of image annotation, it is necessary to minimize the amount of user interaction in this process as much as possible. After investigation, we found that the current interactive models [9,10,11,12,13] perform weakly on the object boundaries and small objects, resulting in the need for a large amount of user interaction in these cases to acquire good segmentation results. The stronger the capability of the interactive segmentation model, the less interaction the user needs to provide each time an image is annotated. To this end, an interactive image-annotation method based on DeepLabv3+ [14] is proposed in this paper to address the dilemma. Furthermore, we propose a layer-by-layer multi-head feature-fusion (MHFF) method and an upsampling attention (UA) mechanism to improve the segmentation ability of the current interactive image-segmentation models.

The contributions of this paper are three-fold:A novel interactive segmentation network for interactive dairy goat image segmentation is proposed, which can significantly lower the time for pixel-level image annotation.A layer-by-layer multi-head feature-fusion method and an upsampling attention mechanism are proposed to enhance the segmentation power of the interactive segmentation model.An interactive dairy goat image-annotation system based on our proposed interactive segmentation model is designed and developed, which only takes 7.12 s on average to annotate a dairy goat instance, five times faster than Labelme.

## 2. Related Work

Interactive image segmentation allows users to guide a network to perform image segmentation by clicks, boxes, and scribble [15], aiming to segment the object of interest in the image with minimum user interaction.

In the early years, traditional interactive image-segmentation methods performed image segmentation via hand-crafted features, such as Graph Cut-based methods [16,17,18], Random Walks-based methods [19,20] and other methods based on regions [21,22,23,24] and contours [25,26,27]. However, traditional methods are mainly based on color, texture, and other hand-crafted features for image segmentation, which are not competent for image segmentation tasks in complex and changeable environments. Convolutional Neural Networks (CNNs) have the power to perceive complex global and local features and have achieved great success in image classification [28,29,30,31,32], object detection [33,34,35,36], semantic segmentation [14,37,38,39,40,41], instance segmentation [42,43,44], and other computer vision tasks [45,46,47]. In recent years, with the popularity of deep learning, a growing number of researchers have tried to employ CNNs in interactive image segmentation.

Xu et al. [10] first proposed a CNN-based interactive segmentation model, which encodes user-provided priors as distance maps and then concatenates them with the original image as its input, achieving a very decent segmentation accuracy compared with traditional methods. Afterward, Liew et al. [11] proposed a region-based method RIS-Net for local refinement by adequately exploiting the context information around the clicks. Maninis et al. [9] used four extreme clicks about the object of interest for image segmentation, lowering the difficulty for users to draw a rectangular box that closely encloses the object of interest when using the interaction method of boxes. Lin et al. [15] proposed a first-click attention mechanism to let the interactive segmentation model better utilize the positional guidance of the first click for the object of interest. In the same year, Zhang et al. [13] proposed a novel interactive segmentation method based on one inside click and two outside clicks.

Despite the good performance of the interactive segmentation model discussed above, a closer inspection reveals that these models have a lower segmentation accuracy for the object boundaries and small objects, which indicates that there is still much room for improvement in the segmentation abilities of current models.

## 3. Material and Methods

The experimental procedures were conducted in accordance with the Guidelines for Animal Experiments set by the Committee for the Ethics on Animal Care and Experiments at Northwest A&F University. Additionally, they were performed under the supervision of the “Guidelines on Ethical Treatment of Experimental Animals” (2006) No. 398 established by the Ministry of Science and Technology, Beijing, China.

### 3.1. Introduction of Dairy Goat Farm Environment

The dairy goat images used for training and testing interactive image-segmentation models in this study were all collected from the dairy goat farm of the Animal Husbandry Teaching and Experimental Base of Northwest A&F University. The dairy goat farm is in Yangling District, Xianyang City, Shaanxi Province, China, at 109.94 east longitude and 34.82 north latitude, and has 149 lactating Saanen dairy goats aged 2–5 years. As shown in Figure 3, it is divided into two scenes: an indoor stable and an outdoor pen. The indoor stable is about 25 m long, 8 m wide and 5 m high. There is an aisle in the middle of the indoor stable and some dairy goat pens divided by iron fences on both sides. Each indoor pen has a small door leading to the outside pen, and there are three windows above each small door to ensure the dairy goat pen has good lighting. The outdoor pen is around 25 m long and 5 m wide. It is a closed area surrounded by an exterior wall of the indoor stable with a height of about 5 m and three hollow brick walls with a height of about 1.2 m. There is a trough inside the outdoor pen for the dairy goats to eat.

### 3.2. Data Collection and Cleaning

In the process of data collection, the initial dairy goat video/image data are mainly collected through fixed cameras and mobile phones.

For the collection method of fixed cameras, the process is as follows: Fixed cameras (Hikvision DS-IPC-B12-I/POE) were set up in the indoor stable and outdoor pen to shoot dairy goat videos, respectively. The shooting specification is 25 fps, 1920×1080 resolution high-definition video, and the shooting time is from 10:00 to 17:00 every day in November 2016. First, in order to increase the diversity of the images in the dairy goat image dataset to be built, we selected video clips from three time periods: morning (10:00–10:30), noon (12:00–12:30), and evening (16:30–17:00) as the candidate video data of dairy goats. After removing the video clips that do not contain dairy goats and dairy goats resting or sleeping all the way, a total of 50 valid clips were obtained. Then, we used FFMpeg to extract key frames from valid dairy goat video clips every 10 s and obtained a total of 9000 frames of dairy goat images. Finally, 1829 dairy goat images were obtained, and the details are shown in Table 1.

As can be seen above, the dairy goat video data were collected in the winter of 2016. However, the data are not sufficient to meet our research needs, so the dairy goat image data were collected in the summer of 2022 by mobile phones to expand the capacity and image diversity of the dataset.

For the collection method of mobile phones, the process is as follows: A mobile phone (Xiaomi 8) was used to take images of dairy goats on 31 July 2022. The image resolution is 2248×1080, and only one dairy goat is included in each image. To boost the variability of the images, we chose two shooting environments of an indoor stable and an outdoor pen, and two postures of standing and laying down and then obtained 1441 dairy goat images. After manually removing some blurry, low-quality, and other occluded images, a total of 1179 dairy goat images were finally obtained. The details are shown in Table 2.

### 3.3. Dataset Annotation and Division

Through the aforementioned two data collection methods, we constructed a dairy goat image dataset containing 3008 images, which we named DGImgs. Then, we used Baidu EasyData to annotate the images in DGImgs, and the total time required is six people/day. Table 3 shows the number of dairy goat instances contained in each image of DGImgs.

According to a ratio of 8:2, DGImgs was divided into training and validation sets, which contain 2407 and 601 images, respectively. Due to the multiple dairy goat instances that may be contained in an image, we can obtain 4319 and 1114 instance-level images from the training set and validation set for model training and validation, respectively.

### 3.4. Proposed Method

In this subsection, we elaborate on our improved interactive segmentation model based on DeepLabv3+.

To reduce the annotation workload of users as much as possible, it is necessary to improve the segmentation power of the interactive segmentation model. To this end, an interactive segmentation network based on DeepLabv3+ is proposed in this paper. To be specific, we proposed a layer-by-layer multi-head feature-fusion (MHFF) method and an upsampling attention (UA) mechanism to improve DeepLabv3+, to boost its segmentation performance on the object boundaries and small objects. We dubbed the improved DeepLabv3+ as UA-MHFF-DeepLabv3+, and its overall architecture is shown in Figure 4.

#### 3.4.1. Interactive Image-Annotation Process

The interactive annotation method includes the following steps.
(1)The user needs to provide some clicks as priors to guide the interactive segmentation network for image segmentation. Specifically, in an image, the pixels in the object to be annotated are the foreground, otherwise the background. Positive and negative clicks are the clicks in the foreground and background, and are used to indicate to the interactive segmentation model that the current pixel belongs to the foreground or background, respectively. As shown in Figure 2, the red dots and green dots in the input image represent the positive clicks and negative clicks provided by the user, respectively.(2)The positive and negative clicks provided by users are encoded into distance maps of positive clicks and negative clicks, respectively. Specifically, the clicks will be translated into distance maps of clicks that have the same width and height as the input image. We use *D* to represent the distance map of positive (or negative) clicks. *H* and *W* represent the height and width of the input image. Then, the distance map can be calculated according to Equation (Equation 1)
(1)$$D\left(i,j\right)=\min_{\forall p_{s}\in S}d\left(p,p_{s}\right)$$0≤i<H and 0≤j<W, D(i,j) is the value of distance map *D* at the index of (i,j), *S* represents the positive (or negative) clicks set, $d(p,p_{s})$ represents the Euclidean distance between the point $p_{s}$ and the point *p*.(3)The input image is first concatenated with distance maps of positive and negative clicks, then input into the interactive segmentation network.(4)If the segmentation accuracy is qualified, the annotation process is over; otherwise, the user needs to provide extra clicks to correct the incorrectly segmented area until the accuracy is qualified or the number of clicks reaches the maximum limit of 20.

#### 3.4.2. Network Architecture

As shown in Figure 4, UA-MHFF-DeepLabv3+ adopts an encoder–decoder architecture. Specifically, the encoder includes the backbone Xception+ and the Atrous Spatial Pyramid Pooling module (ASPP), while the decoder includes the layer-by-layer multi-head feature-fusion module (MHFF) and upsampling attention module (UA), which will be elaborated on in Section 3.4.3 and Section 3.4.4, respectively.

For Xception+, it takes the concatenation of the original image, distance map of positive clicks and negative clicks as input, and then performs image feature extraction from low level (i.e., 2×) to high level (i.e., 16×). The resulting output of 16x downsampling will be fed into the ASPP module for multi-scale context extraction and aggregation. Moreover, before input into the decoder, the feature maps of 2×, 4×, and 8× downsampling will be carried out, providing a channel reduction from 128, 256, and 728 channels to 24, 48, and 144 channels, respectively. The ASPP module consists of a global average pooling operation and four dilated convolutions with dilated rates of 1, 6, 12, and 18, which are capable of extracting image features with a larger receptive field while not lowering the resolution of the input feature maps.

#### 3.4.3. Layer-by-Layer Multi-Head Feature Fusion

In the architecture of the encoder–decoder, the encoder is responsible for extracting image features from low-level to high-level, while the decoder is responsible for fusing the high-level features and low-level features from the encoder and outputting the final segmentation mask. The high-level features usually contain semantic information, such as what the object is, while the low-level features usually contain rich spatial information, such as the object’s location. Only by fully fusing the high-level features and low-level features can the model obtain a decent segmentation accuracy. To this end, we propose a layer-by-layer multi-head feature-fusion method to utilize the low-level features in the encoder fully.

Specifically, the layer-by-layer multi-head feature-fusion method includes the layer-by-layer skip connection feature-fusion structure and the multi-head feature-fusion structure, as shown in Figure 5 and Figure 6, respectively. The layer-by-layer skip connection feature fusion performs layer-by-layer feature fusion from the high level to the low level, and the multi-head feature fusion is responsible for mapping features in two different feature spaces into the same feature spaces.

During the upsampling process of the decoder, the most common feature-fusion method is first to upsample the high-level feature maps to the same resolution as the low-level feature maps, then concatenate them together, and finally perform channel adjustment and feature fusion via a 3×3 convolution kernel. Due to the semantic information in the high-level feature maps and low-level feature maps being different and belonging to different feature spaces, it is not rational to concatenate the two feature maps directly, which may lead to the model’s insufficient learning of the semantic information in high-level and low-level feature maps. Moreover, using the same convolution kernel to map the features in two different feature spaces may lead the model to focus on the learning at one level while omitting the learning at the other level. Therefore, we use two different 3×3 convolution kernels to perform feature space mapping, which are responsible for mapping high-level feature maps and low-level feature maps into the same feature space, respectively. The detailed application process of the multi-head feature fusion is shown in Algorithm 1.
**Algorithm 1** The application process of the multi-head feature fusion**Input:** The high-level feature maps of shape $H\times W\times C^{l}$, the low-level feature maps of shape $2H\times 2W\times C^{h}$**Output:** The fused feature map of shape 2H×2W×48
 1:Perform feature space mapping on the high-level feature maps using a $3\times 3\times C^{l}\times 48$ convolution kernel 2:Perform feature space mapping on the low-level feature maps using a $3\times 3\times C^{h}\times 48$ convolution kernel 3:Upsample 2× the feature maps obtained in step 1 4:Concatenate the feature maps in step 2 and step 3 5:Perform channel adjustment and feature fusion using a 3×3×96×48 convolution kernel on the feature map obtained in step 4

#### 3.4.4. Upsampling Attention

The encoder’s feature-extraction network (i.e., the backbone) extracts features from the image to obtain low-level feature maps and high-level feature maps, which have low-level semantic information and high-level semantic information, respectively. The high-level semantic information is obtained by further feature extraction and abstraction of low-level semantic information, and it has the function of guiding and screening low-level semantic information. The lack of guidance and screening of low-level semantic information by high-level semantic information will limit the feature extraction and expression capabilities. Therefore, to more fully utilize the guidance and screening effect of high-level semantic information on low-level semantic information, we propose an upsampling attention mechanism, which is shown in Figure 7. As can be seen from Figure 7, the proposed upsampling mechanism includes three procedures, namely squeeze, excitation, and scale, which is inspired by the channel attention mechanism in SENet [48] and CBAM [49].
(1)Squeeze: Perform global average pooling and global max pooling on the high-level feature maps $S_{H}$ to obtain two feature vectors of shape $1\times 1\times C^{h}$, respectively.(2)Excitation: Perform nonlinear transformations on the two resulting feature vectors with two sequential fully connected layers to obtain two intermediate weight vectors of shape $1\times 1\times C^{l}$, then perform matrix addition and sigmoid operation on the two intermediate weight vectors to obtain the final weight vector of shape $1\times 1\times C^{l}$, of which each value represents the weight of low-level feature maps.(3)Scale: Use the $1\times 1\times C^{l}$ weight vector to weight each channel of the low-level feature map.

Let *P* represent the global pooling operation, including global average pooling and global max pooling, and *V* represent the nonlinear transformation operation, including the fully connected layer 1, the ReLU function, the fully connected layer 2, the sigmoid function, and Q represent the matrix dot product. Then, the application process of the upsampling attention above can be expressed as shown in Equations (Equation 2) and (Equation 3).
(2)$${\tilde{S}}_{L}=Q(S_{L},V\left(P\left(S_{H}\right)\right)$$
(3)V(x1,x2)=Sigmoid(FC2(ReLU(FC1(x1)))=⊕FC2(ReLU(FC1(x1))))
where $S_{L}$, $S_{H}$, and ${\tilde{S}}_{L}$ represent the low-level feature maps, the high-level feature maps, and the low-level feature maps after channel weighting, respectively. FC1 and FC2 represent the fully connected layer 1 and the fully connected layer 2, which have $C^{h}/r$ and $C^{l}$ neutrons, respectively. ⊕ represents matrix addition. The parameter *r* is used to control the computational complexity of the upsampling attention module; usually, its value is 16.

## 4. Results

### 4.1. Training Settings

The configuration of the deep-learning server utilized for both model training and validation is depicted in Table 4. For all conducted experiments, the training set and validation set derived from our constructed DGImgs dataset were employed for model training and performance evaluation, respectively. The hyperparameter settings are detailed as follows: the cross-entropy loss function was employed, and the stochastic gradient descent optimizer (SGD) with a momentum factor of 0.9 was utilized. The batch size was set to 8, and the iteration period spanned 33 epochs. In the initial 30 epochs, a polynomial learning rate decay strategy (Poly) was employed, followed by the utilization of a constant learning rate for the final three epochs. Prior to being fed into the network, the images were scaled while maintaining their original aspect ratio until the shorter side reached a size of 512 pixels. Subsequently, random image patches of size 512 × 512 pixels were extracted as inputs for the network through the process of random cropping.

### 4.2. Evaluation Metrics

We adopt the mean Number of Clicks (mNoC) as the evaluation metric of the segmentation ability of the interactive segmentation models. mNoC reflects the average number of clicks a user needs to provide to reach a mIoU (mean Intersection over Union) threshold on the dataset. The smaller the mNoC, the better the segmentation ability of the model. Specifically, in the experiments, we employ mNoC@85 and mNoC@90 as the evaluation metrics of all model segmentation capabilities, which represent the average number of clicks a user needs to provide on each sample of the dataset to obtain a mIoU of 0.85 and 0.90, respectively. Please note that if the number of clicks the user provides reaches the maximum limit of 20 and the image-segmentation mask output by the model still cannot obtain the mIoU threshold, the mNoc of the sample will be recorded as 20.

### 4.3. User Clicks Generation

Due to the lack of user clicks in DGImgs, during model training and validation, we employ some user click-generation strategies to simulate the clicks provided by users, including the positive clicks generation strategy and negative clicks generation strategy.

Here, we define $G_{p}$, $G_{n}$, $A^{*}$ and $B^{*}$ to represent the foreground pixels set, the background pixels set, the positive clicks set and the negative clicks set, respectively. Moreover, we use ϕ(p,S) to represent the shortest distance from the point *p* to another region *S*.

**Positive clicks generation strategy.** The generation of positive clicks needs to satisfy the following requirements.
(1)The number of positive clicks needs to be between [1,10].(2)The positive clicks should be the pixels in the foreground, which are $P_{1}$ pixels far away from the background pixels and $P_{2}$ pixels far away from other positive clicks.(3)A new positive click comes from a candidate point set $C_{p}$, as formulated in Equation (Equation 4).
(4)$$C_{p}=\left\{p\in G_{p}|\phi \left(p,G_{n}\right)>P_{1},\phi \left(p,A^{*}\right)>P_{2}\right\}$$

**Negative click-generation strategy.** The generation of negative clicks needs to satisfy the following requirements.
(1)The number of negative clicks needs to be between [0,10].(2)The negative clicks should be the pixels in the background, which are $N_{1}∼N_{2}$ pixels far away from the foreground pixels and $N_{3}$ pixels far away from other negative clicks.(3)A new negative click comes from a candidate point set $C_{n}$, as formulated in Equation (Equation 5).
(5)$$C_{n}=\left\{p\in G_{n}|\phi \left(p,G_{p}\right)\in \left(N_{1},N_{2}\right),\phi \left(p,B^{*}\right)>N_{3}\right\}$$

During model training and validation, the value ranges of $P_{1}$, $P_{2}$, $N_{1}$, $N_{2}$ and $N_{3}$ are {5,10,15,20}, {7,10,20}, {15,40,60}, {80} and {10,15,25}, respectively.

### 4.4. Comparisons with Different Interactive Segmentation Models

To evaluate the difference in the segmentation ability of the existing interactive segmentation model on the DGImgs dataset, we first constructed RIS-Net, DEXTR, IOG, FCANet, and UA-MHFF-DeepLabv3+, and then tested their performance on 601 images in the validation set of DGImgs.

As can be seen from Table 5, the results obtained by UA-MHFF-DeepLabv3+ demonstrate notable improvements in performance, as demonstrated by achieving 1.87 and 4.11 on mNoC@85 and mNoC@90, respectively. These results are notably lower than the previously established state-of-the-art results of 2.73 and 5. Specifically, when compared with IOG, UA-MHFF-DeepLabv3+ exhibited reductions of 1.13 and 0.89 on mNoC@85 and mNoC@90, respectively. Moreover, in comparison with FCANet, UA-MHFF-DeepLabv3+ showcased decreases of 0.86 and 2.48 on mNoC@85 and mNoC@90, respectively. These findings unequivocally underscore the superiority of our proposed model and the indispensable nature of this study.

### 4.5. Ablation Study

To verify the effectiveness of our proposed layer-by-layer multi-head feature-fusion method and upsampling attention, we performed ablation experiments on the validation set of DGImgs. Specifically, we took the DeepLabv3+ as the baseline and then gradually added the MHFF module and the UA module on the DeepLabv3+, and finally tested the segmentation performance of the model after adding each module on 601 images on the validation set. The ablation experiment results are shown in Table 6.

As can be seen, the results in Table 6 reveal the performance gains achieved by incorporating different modules into the baseline DeepLabv3+. To be specific, when compared with the baseline, solely adding the MHFF module, reductions of 0.38 and 0.33 were observed for mNoC@85 and mNoC@90, respectively. Similarly, with the addition of the UA module, decreases of 0.31 and 0.38 were observed for mNoC@85 and mNoC@90, respectively. Notably, the simultaneous incorporation of both the MHFF and UA modules resulted in even greater improvements, with mNoC@85 and mNoC@90 decreasing by 0.64 and 0.62, respectively. The above experimental findings fully demonstrate that the inclusion of either the MHFF module or the UA module consistently enhances the segmentation performance. Moreover, the integration of both modules maximizes the segmentation performance of the model. This is in line with our initial expectations and confirms that the layer-by-layer multi-head feature fusion and upsampling attention introduced in the current interactive model effectively enhance the feature extraction and expression capabilities. As a consequence, users can achieve comparable segmentation accuracy with our proposed UA-MHFF-DeepLabve+ while providing fewer user clicks compared with the previous models.

### 4.6. Visualization of Interactive Dairy Goat Image Segmentation

To illustrate the enhancement effect of the layer-by-layer multi-head feature fusion more vividly and the upsampling attention mechanism on model image feature learning and expression ability, we show the segmentation results of the DeepLabv3+, MHFF-DeepLabv3+, and UA-MHFF-DeepLabv3+ on the DGImgs in the Figure 8.

As can be seen, for the same object to be segmented in the same image at the same click position, both the MHFF-DeepLabv3+ and UA-MHFF-DeepLabv3+ can obtain a finer object boundary and higher accuracy of small object segmentation results than their counterparts. The visualization result of the three models fully demonstrates that our proposed MHFF method and UA mechanism efficiently improve the segmentation ability of existing models and boost their segmentation performance on object boundaries and small objects.

### 4.7. Design and Development of Dairy Goat Image-Annotation System

To promote the implementation of the proposed algorithm, we designed and developed a dairy goat image-annotation system called DGAnnotation for pixel-level annotation of dairy goat images based on UA-MHFF-DeepLabv3+. DGAnnotation is compatible with the Windows operating system (Windows 7 or later) and can be installed to facilitate dairy goat image-annotation tasks performed by livestock husbandry professionals. In addition, we do not have any specific requirements for the observer software’s compatibility. In other words, it is compatible with currently available observer software in the market.

The main interface of the DGAnnotation is shown in Figure 9. It includes eight functions, including opening a folder, finding the previous image and the next image of the currently annotated image, saving an instance, saving all instances in an image, clearing annotations, displaying a file list, and allowing users to annotate images through an interactive annotation panel, corresponding to six buttons of “Open Folder”, “Prev”, “Next”, “Save”, “Save All” and two controls of “Filelist” and “Interactive Annotation Panel” (i.e., the rectangular area where the dairy goat image is located), respectively. With the interactive annotation panel, users can iteratively annotate a dairy goat image in the form of clicks (positive and negative clicks are provided by left and right mouse clicks, respectively).

To evaluate the image labeling efficiency of the DGAnnotation, we first randomly selected 100 dairy goat images from the validation set of the DGImgs and then selected five people to annotate these dairy goat images with DGAnnotation and Labelme, respectively. Finally, we recorded the average time spent by five people on the annotation of the 100 images. The statistical results show that it takes 7.12 s on average to annotate a dairy goat instance using DGAnnotation, while it costs an average of 36.34 s with Labelme. It can be seen that the image-annotation speed of the dairy goat image-annotation system is five times that of Labelme, which fully reflects the high efficiency of our developed dairy goat image-annotation system.

## 5. Discussion

This paper proposes an interactive segmentation model UA-MHFF-DeepLabv3+ for dairy goat image segmentation and annotation, which achieves 1.87 and 4.11 on mNoC@85 and mNoC@90, respectively, as shown in Table 5. These results are notably lower than the previously established state-of-the-art results of 2.73 and 5. Specifically, when compared with IOG, UA-MHFF-DeepLabv3+ exhibited reductions of 1.13 and 0.89 on mNoC@85 and mNoC@90, respectively. Moreover, in comparison with FCANet, UA-MHFF-DeepLabv3+ showcased decreases of 0.86 and 2.48 on mNoC@85 and mNoC@90, respectively. The performance improvement is mainly from the following aspects: (1) The layer-by-layer multi-head feature-fusion method fully utilizes the low-level feature in the encoder to refine the final mask. Therefore, the model consequently has a stronger segmentation capability. (2) The upsampling attention realizes the guidance and screening effect of high-level semantic information on the semantic information, which boosts the feature extraction and expression power of the model. By employing the interactive segmentation model with stronger segmentation abilities, users therefore just need to provide less interaction (in the form of clicks) to reach the same segmentation accuracy as before. To promote the application of the proposed interactive segmentation model, we have developed a dairy goat image-annotation system called DGAnnotation for the pixel-level annotation of dairy goat images. After the test, we found that it takes just 7.12 s to annotate a dairy goat instance with our developed DGAnnotation, five times faster than Labelme. We believe that our research has significant implications for the field of precision livestock farming, as it provides an innovative solution for efficient interactive dairy goat image segmentation and annotation, which in turn facilitates improved monitoring, health assessment, and decision-making in dairy goat farming practices.

Despite the significant progress achieved in our research, it is important to acknowledge the limitations of work and outline the directions for improvement in the future. (1) As can be seen from Table 5, even with our proposed UA-MHFF-DeepLabv3+, it still takes more than four clicks to reach an ideal segmentation performance, which indicates the segmentation capability of the segmentation model still has some margin for improvement. (2) This paper mainly focuses on the single-category task of the dairy goat image annotation. Therefore, in future research, more phenotypical image data, such as sheep images, should be added to the dataset to increase the generalization of the interactive segmentation model.

## 6. Conclusions

To reduce the annotation workload of dairy goat images, we propose a novel interactive segmentation model called UA-MHFF-DeepLabv3+, which employs layer-by-layer multi-head feature fusion and upsampling attention to improve the segmentation accuracy of the DeepLabv3+ on object boundaries and small objects. Experimental results show that our proposed model achieved state-of-the-art segmentation accuracy on the validation set of DGImgs compared with four previous state-of-the-art interactive segmentation models, and obtained 1.87 and 4.11 on mNoC@85 and mNoC@90, which are significantly lower than the best performance of the previous models of 3 and 5. Furthermore, to promote the implementation of our proposed algorithm, we design and develop a dairy goat image-annotation system named DGAnnotation, which can be installed on the Windows operating system (Windows 7 or later) by livestock husbandry technicians for pixel-level annotation of dairy goat images. We demonstrate that it just takes 7.12 s to annotate a dairy goat instance with our developed DGAnnotation, which is five times faster than Labelme. The practical benefits of our research are extensive. By providing a more efficient image-annotation tool, we can reduce the workload of user (i.e., the technicians in the livestock farms) annotation of dairy goat images and facilitate the future study and application of segmentation-based algorithms in the intelligent video surveillance of dairy goats, automatic live body weight estimation and real-time growth monitoring of dairy goats. The development and application of these algorithms offer farmers valuable information about dairy goats, further enhancing their ability to make informed decisions and improve overall farming practices.

## Figures and Tables

**Figure 1 animals-13-03250-f001:**
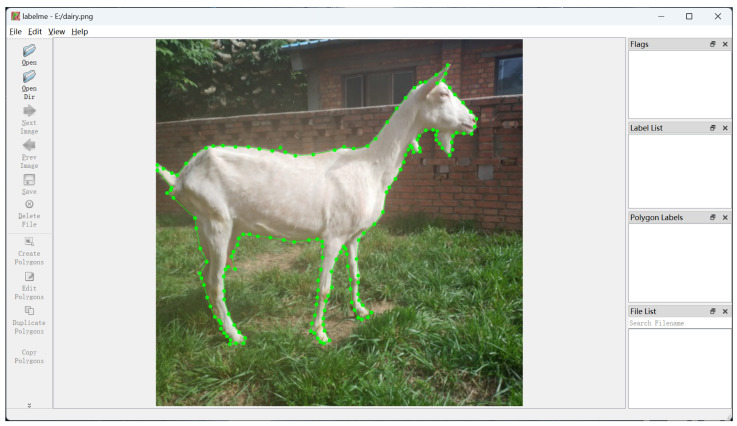
Illustration of the process of the image annotation with Labelme. The green dots in the image are the clicks provided by users to annotate the instance, which is obviously too inefficient and time-consuming to obtain a high-quality annotation result.

**Figure 2 animals-13-03250-f002:**
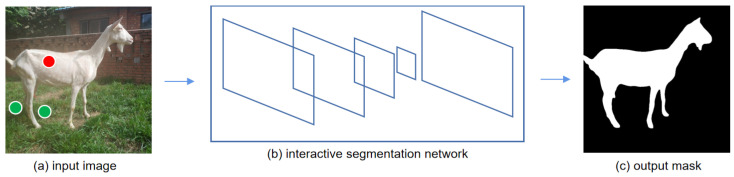
Illustration of the process of the interactive image-annotation method. The red dots and green dots in the input image are positive clicks and negative clicks, respectively, which are user-provided priors for the interactive segmentation network, indicating that the current pixel is the foreground pixel and background pixel, respectively.

**Figure 3 animals-13-03250-f003:**
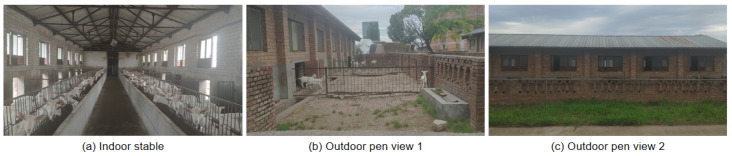
Environment of the dairy goat farm at Animal Husbandry Teaching and Experimental Base at Northwest A&F University.

**Figure 4 animals-13-03250-f004:**
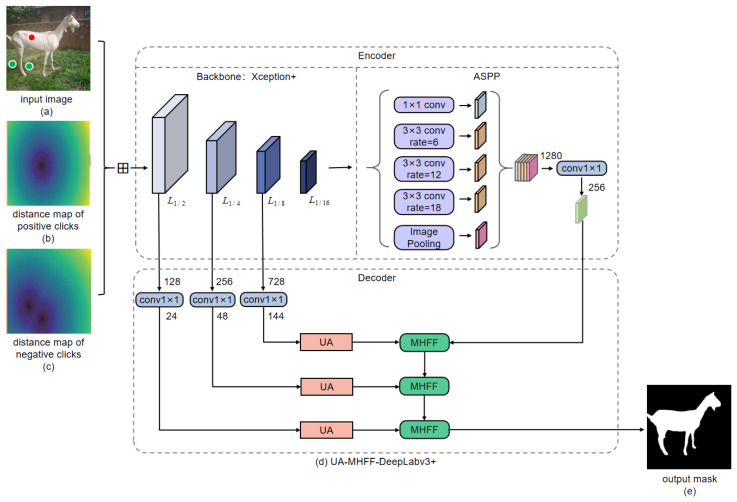
The overall architecture of UA-MHFF-DeepLabv3+. The “UA” and “MHFF” in the decoder represent the upsampling attention mechanism and the multi-head feature-fusion method, respectively. Symbol “⊞” represents the concatenation operation. The red dots and green dots in the input image are the positive clicks and negative clicks provided by the user, respectively.

**Figure 5 animals-13-03250-f005:**
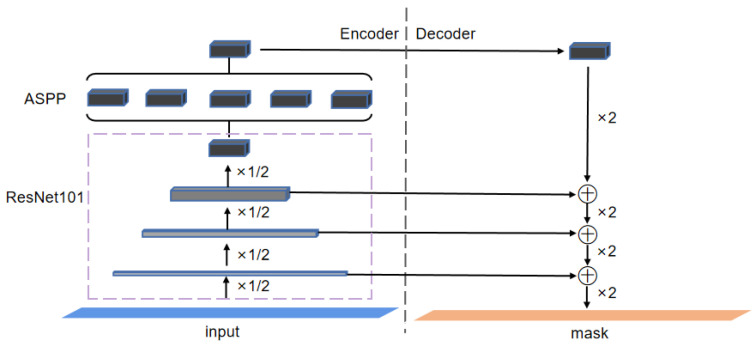
The structure of the layer-by-layer skip connection feature-fusion method.

**Figure 6 animals-13-03250-f006:**
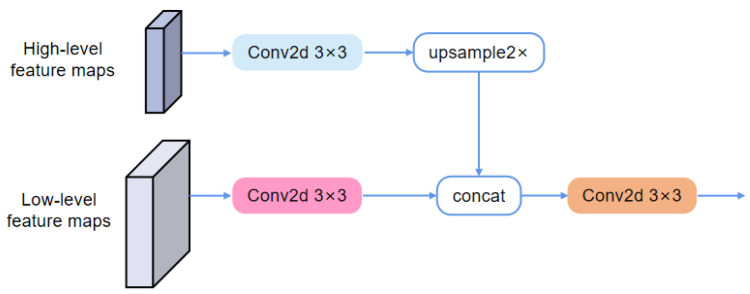
The structure of the multi-head feature-fusion method.

**Figure 7 animals-13-03250-f007:**
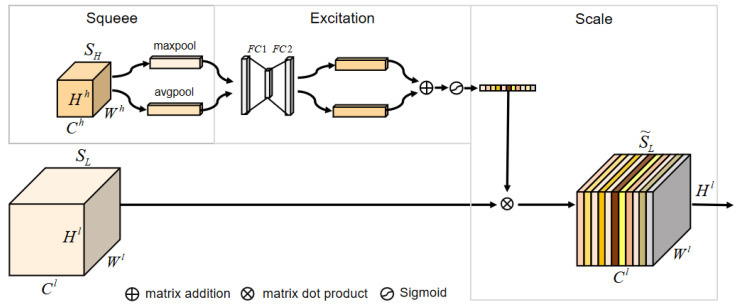
The pipeline of the upsampling attention mechanism. $S_{H}$, $S_{L}$, and ${\tilde{S}}_{L}$ denote the high-level feature maps, the low-level feature maps, and the low-level feature maps after channel weighting, respectively.

**Figure 8 animals-13-03250-f008:**
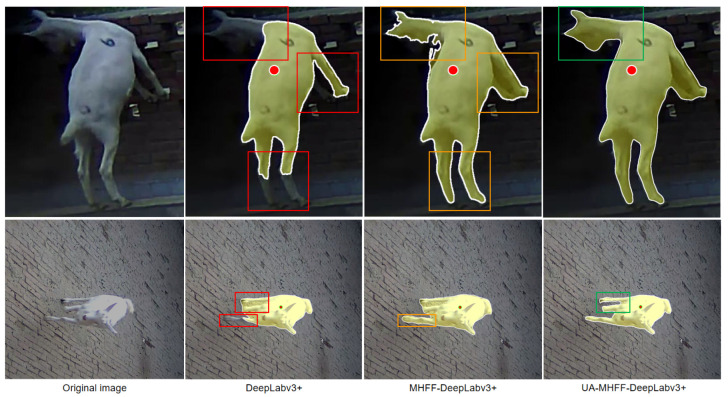
Visualization of the segmentation results of three interactive segmentation models. The red dots are the positive clicks provided by the user. The regions with low segmentation accuracy are marked by red boxes. The improved segmentation regions are marked by orange and green boxes, respectively.

**Figure 9 animals-13-03250-f009:**
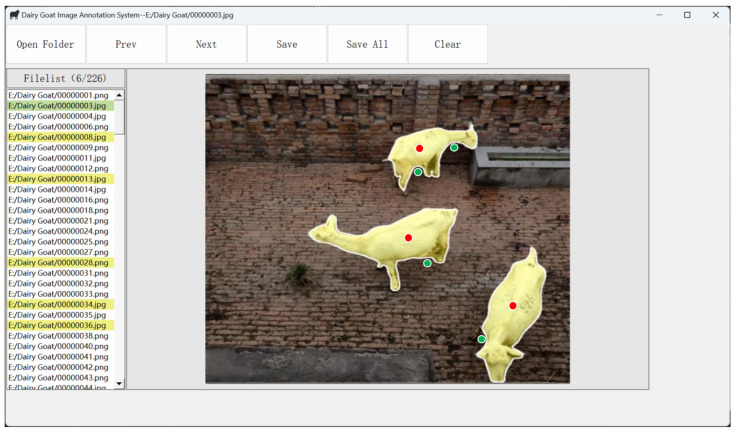
The main interface of DGAnnotation. Six buttons of “Open Folder”, “Prev”, “Next”, “Save” and “Save All” correspond to the functions of opening a folder, finding the previous image and the next image of the currently annotated image, saving an instance, saving all instances in an image, clearing annotations, respectively. Two controls of “Filelist” and “Interactive Annotation Panel” (the rectangular area where the dairy goat image is located) correspond to the functions of displaying a file list and allowing users to annotate images through an interactive annotation panel, respectively. The red and green dots are the positive and negative clicks provided by the user.

**Table 1 animals-13-03250-t001:** Number of dairy goat images captured by fixed cameras.

Time	Environment		In Total
	Indoor	Outdoor	
Morning (10:00–10:30)	299	332	631
Noon (12:00–12:30)	367	401	768
Evening (16:30–17:00)	186	244	430
In total	852	977	1829

**Table 2 animals-13-03250-t002:** Number of dairy goat images captured by mobile phones.

Postures	Environment		In Total
	Indoor	Outdoor	
Standing	246	366	612
Laying down	306	261	567
In total	552	627	1179

**Table 3 animals-13-03250-t003:** Number of dairy goat instances contained in each image of DGImgs.

Number	Shooting Method		In Total
	Fixed Camera	Mobile Phone	
1	533	1179	1712
2	658	0	658
3	324	0	324
4	201	0	201
5	73	0	73
>5	40	0	40

**Table 4 animals-13-03250-t004:** Configuration of the training and inference server.

Configuration	Value
CPU	Intel(R) Xeon(R) Silver 4210 CPU @ 2.20 GHz
GPU	Nvidia GeForce RTX 2080 Ti [11 GB]
RAM	125 GB
Hard Disk	10 TB
OS	Ubuntu 16.04 (64-bit)
Language	python 3.7
Cuda version	10.2
Framework	Pytorch 1.8

**Table 5 animals-13-03250-t005:** Segmentation performance of 5 interactive image-segmentation models on the validation set. ↓ means the smaller the value of the metric, the better the interactive capability of the model. The best results are displayed in bold.

Model	mNoC@85↓	mNoC@90↓
RIS-Net	5.42	8.13
DEXTR	4	6.27
IOG	3	5
FCANet	2.73	6.59
UA-MHFF-DeepLabv3+(Ours)	**1.87**	**4.11**

**Table 6 animals-13-03250-t006:** The contributions of the MHFF module and the UA module to the segmentation performance gains of our proposed network. ✓ and ✗ represent adding and not adding the corresponding module, respectively. ↓ means the smaller the value of the metric, the better the interactive capability of the model. The best results are displayed in bold.

#	DeepLabv3+	MHFF	UA	mNoC@85↓	mNoC@90↓
1	✓	✗	✗	2.51	4.73
2	✓	✓	✗	2.13	4.40
3	✓	✗	✓	2.20	4.35
4	✓	✓	✓	**1.87**	**4.11**

## Data Availability

The data presented in this study are available upon request from the corresponding author. The data are not publicly available due to the privacy policy of the authors’ institution.

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
