# Peer review of "Interactive Dairy Goat Image Segmentation for Precision Livestock Farming"

_animals, 2023, doi:10.3390/ani13203250_

Round 1

Reviewer 1 Report

Aim of the manuscript is to improve the segmentation accuracy of an image annotation tool in dairy goat keeping which is a very important topic. What is missing from the manuscript is the farmers' point of view. As an animal breeder (and not an IT expert), the reviewer has to represent the animal keepers' aspect, as well. That's why the research design and circumstances have to be clearly identified. On the other hand, it is also needed to describe the benefits of the new model for researchers and farmers, as well.

Detailed comments and recommendations are in the attached document.

A few grammatic and stylistic errors have been found and indicated in the text. The specified sentences should be refined.

Reviewer 2 Report

1. An interactive segmentation network for interactive dairy goat image segmentation is proposed in this paper. Also, more phenotypical image data such as sheep images should be added to the dataset to increase the generalization of the interactive segmentation model. Please describe more clearly which parameters should be adjusted in the interactive segmentation model.

2. The abstract and conclusion sections would be improved with more quantitative data from results.

3. Table 5 shows segmentation performance of 5 interactive image segmentation models on the validation set. Please show the unit for the segmentation performance.

4. Are there any limits or requirements for hardware systems such as CPU and RAM, if the proposed method, an interactive segmentation network, is adopted for deep learning experiments?  

5. It is suggested that authors should combine qualitative and quantitative analysis to reflect the repeatability of the proposed method.
